# Effect of β-Blocker Therapy on the Level of Soluble ST2 Protein in Pediatric Dilated Cardiomyopathy

**DOI:** 10.3390/medicina58101339

**Published:** 2022-09-23

**Authors:** Meng Jiao, Xiaofang Wang, Yongmei Liang, Yifei Yang, Yan Gu, Zhiyuan Wang, Zhenyu Lv, Mei Jin

**Affiliations:** 1Department of Pediatric Heart Center, Beijing Anzhen Hospital, Capital Medical University, Beijing 100029, China; 2Beijing Pediatric Heart Center, Beijing 100029, China

**Keywords:** soluble ST2, β-blocker, pediatric dilated cardiomyopathy, adverse events, survival analysis

## Abstract

*Background and Objectives***:** A prognosis for kids with pediatric dilated cardiomyopathy (PDCM) is urgently needed to identify high-risk patients. This study aimed to determine the association of levels and soluble suppression of tumorigenicity 2 (sST2) and medical therapy of β-blocker inhibitors with the risk of adverse events in PDCM. *Materials and Methods*: A total of 124 patients with PDCM were enrolled after admission from 2 centers in China and followed up for adverse events (death, cardiac transplantation, and heart-failure-related rehospitalization). Based on a median sST2 level and the usage of β-blocker inhibitors, patients were divided into four groups. The Cox proportional hazard model was used to assess the risk of incident adverse events. *Results*: The median level of sST2 was 23.77 ng/mL, and 53 (42.7%) patients received β-blocker treatment. Over a median follow-up of 678 days, 37 (29.8%) adverse events occurred. Compared with patients with sST2 < median and without β-blocker, patients with sST2 ≥ median and without β-blocker (HR: 7.01; 95% CI: 1.21–40.45), followed by those with sST2 ≥ median and use of β-blocker had the highest risk of adverse events (hazard ratio (HR): 5.51; 95% confidence interval (CI): 1.17–25.84). However, a significant association was not observed in patients with sST2 < median and use of β-blocker. These associations were consistent across different subgroups. *Conclusions*: A higher level of sST2 was associated with a higher risk of adverse events in patients with PDCM, and β-blocker treatment for children with high levels of sST2 can effectively avoid adverse events.

## 1. Introduction

Pediatric dilated cardiomyopathy (PDCM) is the most common primary cardiomyopathy. Approximately 60% of pediatric cardiomyopathy cases involve dilated cardiomyopathy, which is a serious disorder of cardiac muscle that progresses rapidly to death in children [1]. Although outcomes of children with DCM have improved over the past 2 decades, these children remain at a high risk for death [2]. Therefore, it is necessary to identify prognostic factors to distinguish patients at a higher risk for death.

The suppression of tumorigenicity 2 (ST2) protein is a member of the interleukin-1 receptor family with the transmembrane (ST2L) and soluble (sST2) subtypes that play an important role in several diseases [3]. sST2 is mainly secreted by cardiomyocytes when the cells are subjected to biomechanical overload [4]. However, a recent study reported that endothelial cells could also secrete sST2, which was related to diastolic load [5]. Baseline sST2 levels can predict outcomes in acute and chronic heart failure patients [6]. Previous studies found that sST2 is an independent risk factor for adverse events in patients with DCM and acute heart failure in adults [7]. However, pieces of evidence of the effect of sST2 on the prognosis in pediatric patients with DCM remain unknown. Furthermore, it has been demonstrated that PDCM is treated with drugs such as β-blocker inhibitors that may relieve the symptoms of heart failure and maximize cardiac function [8], while whether the use of drugs such as β-blocker inhibitors influences the level of sST2 and the prognosis of PDCM, was not investigated. Therefore, in the present study, we aimed to assess the combined effect of sST2 and the use of β-blocker inhibitors on adverse events in PDCM.

## 2. Materials and Methods

### 2.1. Study Population

This study is a double-center, observational, prospective, integrative omics study aiming to determine the diagnostic and prognostic value of novel biomarkers in cardiomyopathy. The flowchart of the study was presented in Figure 1. A total of 176 DCM children (aged < 18 years) were enrolled in Beijing Anzhen Hospital from September 2015 to March 2017. The exclusion criteria were as follows: presence or history of systemic diseases, such as diabetes, uremia, rheumatic fever, Kawasaki disease, hypertension, or congenital cardiovascular malformations; with a toxic exposure known to cause heart muscle disease (anthracyclines, mediastinal radiation, iron overload, or heavy metal exposure); bedridden for >3 months and/or unable to stand alone (age > 2 years); survival < 30 days; previous history of chronic liver disease or alanine aminotransferase levels > 80U/L; and a previous history of renal dysfunction. Finally, a total of 124 patients were enrolled in the final analysis. All patients had follow-up visits every 3 months. Follow-up data were prospectively obtained from medical records, communication with patients’ physicians, phone follow-ups, and patients’ regular visits to staff physicians at outpatient clinics.

The study was designed and carried out by following the principles of the Declaration of Helsinki. The study was approved by Beijing Anzhen Hospital Ethics Committee, and all informed consent was obtained; the project ID is 2016023, which was approved on 1 March 2020. More details are presented at ClinicalTrials.gov (accessed on 1 March 2020, NCT03076580).

### 2.2. Disease Classification

DCM and heart failure was diagnosed and identified by at least 3 experienced cardiologists. DCM was defined as the presence of 2 or 3 of the following criteria: (1) symptomatic heart failure, (2) left ventricular (LV) or biventricular systolic dysfunction, and (3) dilatation that was not explained by abnormal loading conditions or abnormalities of the coronary arteries. Systolic dysfunction was defined by abnormal LV fractional shortening (25%). LV dilatation was defined by an LV end-diastolic volume or diameter > 2 standard deviations (SDs) from normal according to nomograms (z scores > 2) corrected by body surface area and age [9].

### 2.3. Data Collection and Measurement of sST2

All data were extracted and identified from electronic health records, including demographic information, medical history, and medication on β-blocker. Blood samples were collected from participants with empty stomachs and drawn into sterile polyolefin resin tubes with a coagulant. The samples were then centrifuged at 3000 rpm for 10 min in the clinical laboratory. The supernatant serum was quickly removed, aliquoted, and stored at −80 °C (−112 °F). The serum samples were stored at −80 °C for <2 years before the sST2 levels were analyzed. Laboratory data, transthoracic echocardiography, and sample quantification were recorded and time-stamped, as previously described [1]. Circulating sST2 levels were measured by a high-sensitive sandwich immunoassay (Critical Diagnostics, San Diego, CA, USA).

### 2.4. Statistical Analysis

Participants were classified into 4 categories according to the median sST2 value (23.77 ng/mL) and the use of β-blockers: sST2 < median and without β-blocker use, sST2 ≥ median and without β-blocker use, sST2 < median and with β-blocker use, and sST2 ≥ median and with β-blocker use. Baseline data were described as a median (interquartile range (IQR): Q1–Q3) or as percentages, as appropriate. Differences between groups were evaluated using the Mann–Whitney *U* or Kruskal–Wallis for a continuous variable and chi-square for categorical variables. Kaplan–Meier methods were performed to evaluate the incidence rate of outcomes, and differences among groups were evaluated by a log-rank test. Person-years were calculated from the baseline to the first occurrence of a predefined adverse event (death/heart transplantation/heart-failure-related rehospitalization) or the end of the study, whichever came first. The incidence rate was calculated by dividing the number of incident cases by the total follow-up duration (person-year).

Cox regression analysis was used to determine the association between the four categories and predefined adverse events, taking the group of sST2 < median and without β-blocker as a reference. Variables with *p* < 0.1 and well-established predictors were selected as confounding variables in the multivariable analyses. The proportional hazards assumption was satisfied by checking the Schoenfeld residual plots. Subgroup analyses stratified by variables with *p* < 0.1 in the univariate analysis were also performed, including gender (female vs. male), previous history of heart failure (no vs. yes), septal thickness (<median vs. ≥median), urea (<median vs. ≥median), and BNP (<median vs. ≥median). Interactions between subgroups were tested using the likelihood ratio tests comparing models with and without multiplicative interaction terms. Analyses were performed with SPSS 21.0 (IBM, Chicago, IL, USA) and R version 3.4.0 (R Core Team, Vienna, Austria). All the tests were two-sided, and a *p*-value of < 0.05 was considered statistically significant.

## 3. Results

### 3.1. Baseline Characteristics

Among 124 enrolled patients, the median age was 22.0 months (IQR: 10.5–69.5 months), and 39.8% were female. The overall range of sST2 levels was 13.67 to 104.91 ng/mL, and the median of sST2 levels was 23.76 ng/mL (interquartile range: 19.66 to 30.87 ng/mL). There were 30 (24.47%) patients with sST2 < median and without β-blocker use, 41 (32.98%) with sST2 ≥ median and without β-blocker use, 32 (25.53%) with sST2 < median and with β-blocker use, and 21 (17.02%) with sST2 ≥ median and with β-blocker use. Baseline characteristics among the four groups were presented in Table 1. In general, the baseline characteristics were well balanced among the four groups, except patients with sST2 ≥ median and with β-blocker use had the highest proportion of males (*p* = 0.0205), and sST2 < median and with β-blocker use had the highest prevalence of heart failure (*p* = 0.0253).

During a median follow-up of 678 days (IQR: 533–785 days), 37 patients had all-cause death (n = 10), cardiac transplantation (n = 4), and rehospitalization for heart failure (n = 23). Significantly higher LV dilation and BNP levels were found in patients with adverse events (LV ejection fraction, *p =* 0.0045; LV fraction shortening, *p =* 0.0218; left atrial end-diastolic diameter (LVEDD), *p =* 0.024; BNP, *p =* 0.0003), compared with those without adverse events (Table 2).

### 3.2. The Combined Effect of sST2 and β-Blocker on Adverse Events in PDCM

Cases of adverse events were three (8.70%) for patients in sST2 < median and without β-blocker use group, 16 (38.71%) for patients in ST2 ≥ median and without β-blocker use group, 15 (46.88%) for patients in sST2 < median and without β-blocker use group, and 4 (20.83%) for patients in sST2 ≥ median and with a β-blocker group (Table 3). The highest incidence rate was observed in the sST2 ≥ median and without a β-blocker group (0.97; 95% CI, 0.50–1.86 per 1000-person year). The Kaplan–Meier curve also showed that patients with sST2 ≥median and without β-blocker use experienced the highest risk of adverse events compared with patients in other groups (log-rank test, *p =* 0.021; Figure 2). This trend persisted even after adjusting for age, gender, previous history of heart failure, septal thickness, and urea. Patients in sST2 ≥ median with the β-blocker group and sST2 ≥ median without the β-blocker group had a significantly higher risk of adverse events, where the hazard ratio was 5.51 (95% CI, 1.17–25.84) and 7.01 (95% CI, 1.21–40.45), respectively (Table 3), compared to patients in the sST2 <median and without β-blocker use group.

### 3.3. Subgroup Analysis

Results of the subgroup analysis were presented in Table 4. We found there was no significant interaction between the combined effect of the sST and the β-blocker. Stratified variables in relation to adverse events, indicting the combined effect of the sST and the β-blocker on adverse events, were consistent across different subgroups, including gender (*p* for interaction = 0.5380), previous history of heart failure (*p* for interaction = 0.8702), septal thickness (*p* for interaction = 0.7741), urea (*p* for interaction = 0.8019), and BNP (*p* for interaction = 0.8914).

## 4. Discussion

The patients with PDCM easily obtained dilated cardiomyopathy, and moreover, progressed rapidly to death. Currently, most studies focus on finding prognostic factors of these outcomes to decrease the occurrence of such incidents in adults, but little research on children [10]. In this study, electronic health records data, blood samples, and laboratory data were collected, and pediatric participants were classified into four categories according to the median sST2 value (23.77 ng/mL) and the use of β-blockers. Cox regression analysis was used to determine the association between the four categories and predefined adverse events, which found that PDCM patients with high levels of sST2 were more probably prone to adverse events, no matter the usage of β-blocker therapy or not. In addition, the β-blocker therapy could really decrease the adverse effect of high levels of sST2, but it could not eliminate the adverse effect totally.

In this study, the median sST2 concentration of 23.77 ng/mL was used as the standard to classify high and low concentrations, which is almost the same as other pediatric research. Courtney et al. used the cohort median (23 ng/mL) of day + 14 sST2 as a cut-off for risk categorization into high (above) and low (below) [11]. In addition, we found that patients with high levels of sST2 and without β-blocker use had the highest prevalence of heart failure, which is not difficult to understand. At present, many studies have found that sST2 is a risk factor for heart failure disease, whether it is in adults or children [12,13,14]. In addition, patients with high levels of sST2 are more likely to be recommended to undergo β-blocker treatment to control their condition [13,15,16]. Therefore, the highest level of sST2 and β-blocker treatment group had the most patients with heart failure. More importantly, this shows that the median concentration of sST2 used in this study had a good distinguishing effect on patients.

There are also some interesting findings. We found that children patients in the high level of sST2 with β-blocker use group and the high level of sST2 without β-blocker use group had a significantly higher risk of adverse events, where the hazard ratio was 5.51 (95% CI, 1.17–25.84) and 7.01 (95% CI, 1.21–40.45), respectively, after adjusting for age, gender, previous history of heart failure, septal thickness, and urea. Hanna K et al. conducted a study on whether sST2 and β-blocker treatments in adults with heart failure caused adverse events; they found that those with the highest risk for subsequent cardiovascular events were identified by an elevated baseline sST2 value, but this risk was not entirely realized in those titrated to higher doses of β-blocker [13]. β-blocker treatments could eliminate or weaken the risk effect of sST2. Furthermore, Jianli Bi et al. collected various studies on the risk factors of heart disease in children and concluded that patients with low sST2/high-dose BB had the lowest cardiovascular event rate (0.53 events); those with low sST2/low-dose β-adrenergic blocker, or high sST2/high-dose β-adrenergic blocker had intermediate outcomes (0.92 and 1.19 events); patients with high sST2 treated with low-dose β-adrenergic blocker had the highest cardiovascular event rate (2.08 events) [17]. These examples of evidence also show that β-blocker treatment can indeed reduce the risk of sST2 [18,19,20,21], which is consistent with the results in this study.

This study has some limitations. Firstly, although this study is a dual-center, prospective, observational study from two major heart disease centers in China, this study cannot fully represent the global average DCM population. Due to the large size of the cohort, it is difficult to determine the disease-specific reference range of sST2 and the diagnostic reference range of sST2 for pediatric patients with DCM. Therefore, it is necessary to conduct further large-scale research. Age affects sST2 levels and is an independent risk factor for heart failure [22]. However, in our study, age, gender, previous history of heart failure, septal thickness, and urea have nothing to do with adverse events, and when added as a covariate, it does not affect the hazard ratio. Additionally, some information was not collected in our study, such as acute heart failure stage, sST2 during the follow-up, categories of DCM, and types of beta-blockers, which should be paid more attention to in future investigations. Finally, in the subgroup analysis, we included patients with late adverse events (>6 months), as well as age, gender, and drug-matched controls, but the sample size was limited. A joint model is needed to evaluate the combined effects of sST2 and beta in larger multicenter studies and more frequent measurements.

In this study, the sST2 level of pediatric patients with DCM and the effect of β-blocker treatment on the occurrence of adverse events were conducted. sST2 is a special biomarker that can distinguish patients at a high risk of death from those who may recover, and sST2 has strong prognostic value in short-term and long-term adverse events. In addition, β-blocker treatment for children with high levels of sST2 can effectively avoid adverse events. However, the specific dose level of beta-blocker used that can reduce the risk of adverse events is unknown, and further research is needed in the future.

## Figures and Tables

**Figure 1 medicina-58-01339-f001:**
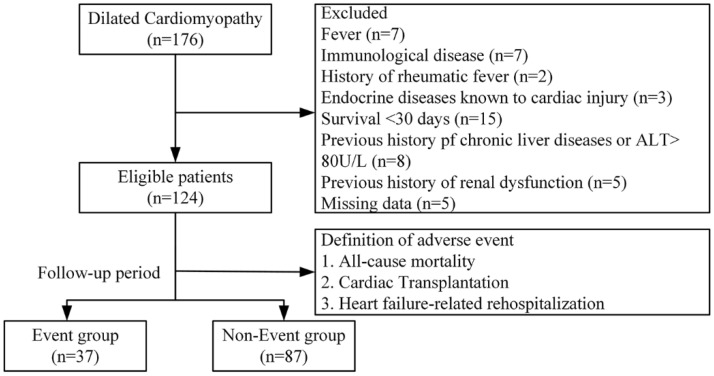
The flowchart of the study. Abbreviation: ALT, Alanine aminotransferase.

**Figure 2 medicina-58-01339-f002:**
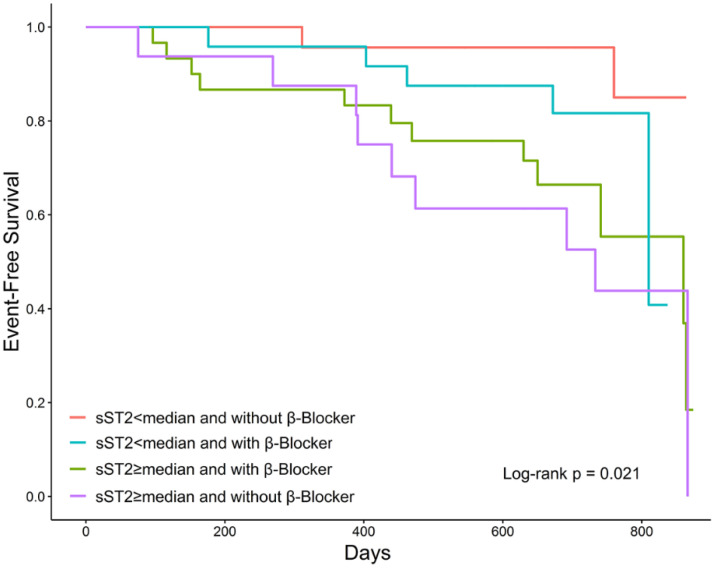
Kaplan–Meier estimate value and the risk of adverse events ranked by the combination of sST2 levels and use of β-blockers. Abbreviation: sST2, soluble suppression of tumorigenicity 2.

**Table 1 medicina-58-01339-t001:** Baseline characteristics according to different sST2 level with or without β-Blocker.

Characteristics	Without β-Blocker		With β-Blocker		*p* Value
	sST2 < Median	sST2 ≥ Median	sST2 < Median	sST2 ≥ Median	
Participants, n (%)	30 (24.47)	41 (32.98)	32 (25.53)	21 (17.02)	
Age, months	17.0 (8.0–48.0)	33.0 (10.0–117.0)	23.0 (15.0–64.5)	52.5 (10.5–107.5)	0.5886
Male, n (%)	13 (39.1)	26 (64.5)	17 (54.2)	18 (87.5)	0.0205 *
Previous history of heart failure, n (%)	9 (30.4)	15 (35.5)	19 (58.3)	3 (12.5)	0.0253 *
LVEF raw	37.0 (34.0–46.0)	37.0 (31.0–41.0)	41.5 (32.0–46.5)	38.0 (32.5–43.0)	0.4403
LV fraction shortening raw	17.0 (16.0–23.0)	18.0 (15.0–20.0)	19.5 (13.0–23.0)	19 (15.5–21.0)	0.8248
LVEDD, z score	4.3 (3.2–5.8)	4.8 (3.3–5.2)	4.9 (3.9–6.5)	3.5 (2.5–5.5)	0.1605
Septal thickness, z score	1.4 (0.7–2.2)	1.1 (0.0–1.8)	0.8 (0.2–1.8)	0.4 (0.04–1.0)	0.0550
LV posterior wall thickness, z score	1.0 (0.5–1.9)	0.8 (0.08–1.4)	1.1 (0.4–1.8)	0.5 (0.08–1.0)	0.3003
Laboratory data					
HDL cholesterol, mmol/L	1.3 (1.1–1.7)	1.3 (1.1–1.5)	1.4 (1.1–1.8)	1.2 (1.1–1.4)	0.6063
LDL cholesterol, mmol/L	2.6 (1.8–3.6)	2.3 (1.8–2.8)	2.5 (2.1–3.2)	2.2 (1.8–2.9)	0.8201
Total cholesterol, mmol/L	4.4 (3.5–5.6)	4.1 (3.4–4.6)	4.3 (3.7–5.2)	3.6 (3.6–4.5)	0.3491
Creatinine, μmol/L	26.5 (21.6–37.5)	27.6 (20.6–37.1)	30.0 (23.5–42.8)	34.9 (22.7–41.1)	0.6145
Urea, μmol/L	3.3 (3.0–4.3)	4.6 (3.8–5.3)	4.5 (3.4–5.9)	3.9 (3.7–4.8)	0.0620
Glucose, mmol/L	5.1 (4.6–5.3)	4.9 (4.6–5.1)	4.9 (4.5–5.3)	4.9 (4.5–5.1)	0.8298
BNP, pg/mL	145.0 (73.0–411.0)	324.0 (91.0–605.0)	242.5 (68.5–495.5)	306.5 (169.5–621.0)	0.4426
C-reactive protein, μmol/L	0.2 (0.1–0.5)	0.3 (0.2–0.6)	0.16 (0.1–1.2)	0.1 (0.1–1.1)	0.6137

Values are represented as median (Q1-Q3). The median of sST2 was 23.77 ng/mL. ACEI, angiotensin-converting enzyme inhibitors; BNP, B-type natriuretic peptide; EDD, end-diastolic dimension; EF, ejection fraction; HDL, high-density lipoprotein; LDL, low-density lipoprotein; LV, left ventricular; sST2, soluble ST2. * *p* < 0.05 is significant.

**Table 2 medicina-58-01339-t002:** Baseline characteristics of pediatric DCM patients with or without adverse events.

Characteristics	Patients without Adverse Events (n = 87)	Patients with Adverse Events (n = 37)	*p* Value
Age, months	24.0 (12.0–59.0)	15.0 (10.0–104.0)	0.5909
Male, n (%)	48 (54.6)	26 (71.4)	0.1272
Previous history of heart failure, n (%)	28 (31.8)	18 (46.4)	0.1776
Echocardiographic measurements at enrollment			
LV ejection fraction raw	41.0 (32.0–46.0)	35.0 (32.5–38.0)	0.0045 *
LV fraction shortening raw	20.0 (15.0–23.0)	17.0 (15.5–19.0)	0.0218 *
LVEDD, z score	3.9 (3.1–5.6)	4.9 (4.4–6.2)	0.0235 *
Septal thickness, z score	1.0 (0.2–1.8)	0.9 (0.1–1.6)	0.9934
LV posterior wall thickness, z score	0.8 (0.2–1.8)	0.7 (0.07–1.3)	0.4492
Laboratory data			
HDL cholesterol, mmol/L	1.3 (1.1–1.6)	1.2 (0.9–1.5)	0.1543
LDL cholesterol, mmol/L	2.3 (1.8–3.0)	2.6 (2.1–3.5)	0.2287
Total cholesterol, mmol/L	4.1 (3.5–5.2)	4.2 (3.7–5.9)	0.2326
Creatinine, μmol/L	27.4 (22.4–39.1)	35.0 (21.3–40.0)	0.5302
Urea, μmol/L	4.0 (3.3–5.2)	4.0 (3.0–5.1)	0.7391
Glucose, mmol/L	4.9 (4.5–5.3)	4.9 (4.5–5.2)	0.9621
BNP, pg/mL	164.0 (69.0–411.0)	473.5 (285.5–602.0)	0.0003 *
C-reactive protein, μmol/L	0.2 (0.1–0.7)	0.3 (0.1–1.2)	0.4163

Values are represented as median (Q1–Q3). ACEI, angiotensin-converting enzyme inhibitors; BNP, B-type natriuretic peptide; DCM, dilated cardiomyopathy; EDD, end-diastolic dimension; EF, ejection fraction; HDL, high-density lipoprotein; LDL, low-density lipoprotein; LV, left ventricular; sST2, soluble ST2. * *p* < 0.05 is significant.

**Table 3 medicina-58-01339-t003:** Association of sST2 levels and β-Blocker treatment with the risk of adverse events.

	Without β-Blocker		With β-Blocker	
	sST2 < Median	sST2 ≥ Median	sST2 < Median	sST2 ≥ Median
Case, n (%)	3 (8.70)	16 (38.71)	15 (46.88)	4 (20.83)
Incidence rate (per 1000-person year)	0.12 (0.03–0.49)	0.69 (0.39–1.21)	0.97 (0.50–1.86)	0.31 (0.13–0.75)
Unadjusted				
HR (95% CI)	Reference	5.74 (1.28–25.85)	7.12 (1.52–33.35)	2.67 (0.52–13.79)
*p*-value		0.0228	0.0128	0.2403
Adjusted				
HR (95% CI)	Reference	5.51 (1.17–25.84)	7.01 (1.21–40.45)	2.47 (0.45–13.69)
*p*-value		0.0305	0.0296	0.3009

CI, confidence interval; HR, hazard ratio; sST2, soluble ST2. The median of sST2 was 23.77 ng/mL. Adjusted for age, gender, previous history of heart failure, septal thickness, and urea.

**Table 4 medicina-58-01339-t004:** Subgroup analyses for the association of sST2 levels and β-Blocker treatment with risk of adverse events.

Variables	Without β-Blocker		With β-Blocker		*P* for Interaction
	sST2 < Median	sST2 ≥ Median	sST2 < Median	sST2 ≥ Median	
Gender					
Female	Reference	11.51 (0.79–166.84)	165.63 (2.98–921.25)	3.13 (0.13–74.63)	0.5380
Male	Reference	2.82 (0.30–26.34)	3.99 (0.41–38.85)	1.87 (0.19–18.70)	
Previous history of heart failure					
No	Reference	2.33 (0.22–25.04)	2.92 (0.26–32.63)	2.37 (0.21–27.21)	0.8702
Yes	Reference	6.15 (0.68–55.41)	8.17 (0.38–177.12)	1.70 (0.14–20.27)	
Septal thickness, z score					
<Median	Reference	3.95 (0.30–52.31)	5.25 (0.47–58.81)	1.13 (0.08–16.04)	0.7741
≥Median	Reference	6.51 (0.61–69.86)	6.87 (0.42–111.05)	2.86 (0.24–33.52)	
Urea, μmol/L					
<Median	Reference	6.96 (0.67–72.33)	10.21 (0.80–130.82)	3.68 (0.30–44.54)	0.8019
≥Median	Reference	1.58 (0.15–16.38)	2.09 (0.13–34.15)	0.36 (0.02–6.20)	
BNP					
<Median	Reference	2.18 (0.04–10.56)	6.17 (0.23–34.57)	1.64 (0.02–11.01)	0.8914
≥Median	Reference	5.20 (0.78–34.81)	5.55 (1.10–28.07)	2.14 (0.34–13.60)	

The median of sST2 was 23.77 ng/mL. Adjusted for age, gender, previous history of heart failure, septal thickness, and urea other than the variable for stratification. sST2, soluble ST2; BNP, B-type natriuretic peptide.

## Data Availability

Data are available to researchers upon request for purposes of reproducing the results or replicating the procedure by directly contacting the corresponding author.

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
