# Peer review of "Effect of β-Blocker Therapy on the Level of Soluble ST2 Protein in Pediatric Dilated Cardiomyopathy"

_medicina, 2022, doi:10.3390/medicina58101339_

Round 1

Reviewer 1 Report

1.      Introduction: “whether the use of drugs such as β-blocker inhibitors influences the level of sST2” Did the authors provide data about it?.

2.      Methods: it is not clear when samples (of sST2) were performed.

3.      Methods: please provide data about the situation of heart failure (acute or chronic, stage etc during the day that measurements of the sST2 was performed.

4.      Are there any data about sST2 levels during the follow up?

5.      Are there any data about other biomarkers

6.      Subgroup analysis: Please provide data regarding the association of Sst2 and b-blocker treatment in patients with/without high BNP levels

7.      What the sST2 really added in heart patients? Provide data regarding the combination of high sST2 levels and use of b-blockers. Is it a different Kaplan-Meier curve?

Author Response

  1. Introduction: “whether the use of drugs such as β-blocker inhibitors influences the level of sST2” Did the authors provide data about it?.

[Response] Thanks for your comment. There was no relevant data on the effect of β-blocker inhibitors on the levels of sST2, thus we demonstrated that “whether the use of drugs such as β-blocker inhibitors influences the level of sST2 and the prognosis of PDCM was not investigated.”

  1. Methods: it is not clear when samples (of sST2) were performed.

[Response] Thanks for your suggestion. The samples of sST2 were collected collected from participants with empty stomachs and drawn into sterile polyolefin resin tubes with coagulant. The samples were then centrifuged at 3000 rpm for 10 minutes in the clinical laboratory. The supernatant serum was quickly removed, aliquoted, and stored at -80℃ (- 112 °F). The serum samples were stored at -80℃ for < 2 years before the sST2 was analyzed. The time when samples were performed were added into the revised manuscript.

  1. Methods: please provide data about the situation of heart failure (acute or chronic, stage etc during the day that measurements of the sST2 was performed.

[Response] Thanks for your comment. It is a good idea to present the situation of heart failure stage during the day that measurements of the sST2. however, we did not collected the relevant information, which is one of the limitations of our study, and we have demonstrated in the discussion section in the revised manuscript.

  1. Are there any data about sST2 levels during the follow up?

[Response] Thanks for your comment. sST2 were only collected at baseline, and we did not collected the information of sST2 levels during the follow up. This limitation has been demonstrated in the revised manuscript.

  1. Are there any data about other biomarkers

[Response] Thanks for your comment. There were data for other biomarkers, such as lipid profiles, creatinine, urea, glucose, B-type natriuretic peptide (BNP), and C-reactive protein (CRP), these data were presented in Table 1.

  1. Subgroup analysis: Please provide data regarding the association of Sst2 and b-blocker treatment in patients with/without high BNP levels

[Response] Thanks for your suggestion. Subgroup analysis were further performed in patients with and without high BNP (median value of 245 pg/mL). Results on the subgroup analysis stratified by BNP levels have been added into the revised manuscript, showing the association of sST2 and b-blocker treatment with the risk of adverse events were consistent across patients with different BNP levels.

  1. What the sST2 really added in heart patients? Provide data regarding the combination of high sST2 levels and use of b-blockers. Is it a different Kaplan-Meier curve?

[Response] Thanks for your suggestion. In our study, the Kaplan-Meier curve was performed by stratified the patients into 4 categories as “sST2 < median and without b-blockers”, “sST2 < median and with b-blockers”, “sST2 ≥ median and without b-blockers”, and “sST2 ≥ median and with b-blockers”. The results showed the risk of adverse events was higher in patients with sST2 ≥median regardless they used or not used b-blockers, indicating a higher sST2 level was strongly associated with the risk of adverse outcomes.

Reviewer 2 Report

Please, rephrase this sentence: "The patients with PDCM (pediatric dilated cardiomyopathy) were easy to get dilated cardiomyopathy".

Was EMB performed? And if it was performed, was there any changes in the causes of heart failure? Sometimes DCM without histological confirmation may include ARVD, non-compacted myocardium or developed after myocarditis. If there was no EMB, please, add more information how patients were considered as having DCM, i.e. only TTE?

Author Response

Please, rephrase this sentence: "The patients with PDCM (pediatric dilated cardiomyopathy) were easy to get dilated cardiomyopathy".

Was EMB performed? And if it was performed, was there any changes in the causes of heart failure? Sometimes DCM without histological confirmation may include ARVD, non-compacted myocardium or developed after myocarditis. If there was no EMB, please, add more information how patients were considered as having DCM, i.e. only TTE?

[Response] Thanks for your comment. EMB was not performed in our current study. DCM and heart failure was diagnosed and identified by at least 3 experienced cardiologists. DCM was defined as the presence of 2 or 3 of the following criteria: (1) symptomatic heart failure, (2) left ventricular (LV) or biventricular systolic dysfunction, and (3) dilatation that was not explained by abnormal loading conditions or abnormalities of the coronary arteries. This has been demonstrated in the “Disease classification” section.

Reviewer 3 Report

The authors studied an efficacy of beta-blockers on pediatric DCM according to a level of the suppression of tumorigenicity-2 (ST2) protein. ST-2 has bee gained interest as a potential biomarker in cardiovascular disease because IL-33 and ST-2 ligands consist of the cardioprotective pathway which inhibits inflammatory responses, following prevention of fibrosis, hypertrophy, and apoptosis of cardiomyocytes.

The authors described that a total of 146 DCM children (aged <18 years) were enrolled in Beijing Anzhen Hospital; however, the number of DCM children was described as 176 in Figure 1. Please confirm the number of enrolled patients.

Abstract:

This study was a double-center, observational, prospective, integrative omics study, aiming to determine the diagnostic and prognostic value of serum sST-2 level in children with DCM. The authors have to clearly describe the outcome of this study in the abstract. I guess that the outcome will seem to the event-free survival.

Introduction:

Introduction was well written.

Methods:

There are several concerns regarding the method of this study. This study aimed to determine the association of serum level of sST-2 and beta-blocker therapy. Therefore, it is required to exactly explore the feasibility and utility of measuring serum levels of sST-2 to predict outcomes for pediatric DCM, and the effects on beta-blocker therapy, i.e. whether beta-blacker therapy could reduce serum levels of sST-2. At first, the authors have to analyze the event-free survival in overall patients, and to identify the factors relevant to the survival, including treatment with a beta-blocker and serum levels of sST-2. Subsequently, if high serum levels of sST-2 would be identified as a poor prognostic factor, the authors have to provide the ROC curve to identify a cut-off value, the sensitivity and specificity of sST-2 levels among children with DCM. Finally, the authors have to study whether beta-blocker therapy contributes to poor outcomes and could reduce serum levels of sST-2.

The authors have to describe the indication of beta-blocker therapy, and types of beta-blockers in the study population.

The authors have to describe the categories of DCM in the study population, including idiopathic, post-myocarditis, ischemic, neuromuscular, and metabolic disorders, since pediatric DCM encompasses the different pathophysiology and predisposing diseases.

Results:

Serum levels of sST-2, including ranges, in overall patients should be described.

The entire of results should be rewritten according to suggestions described as the above.

Discussions

It is postulated that sST-2 is a biomarker in monitoring of heart failure, which is described in the 2017 American College of Cardiology/American Heart Association guidelines. The discussions should be strengthened to focus on the relationship between the pathophysiology of pediatric DCM and chronic inflammatory responses which deteriorates

Author Response

The authors studied an efficacy of beta-blockers on pediatric DCM according to a level of the suppression of tumorigenicity-2 (ST2) protein. ST-2 has bee gained interest as a potential biomarker in cardiovascular disease because IL-33 and ST-2 ligands consist of the cardioprotective pathway which inhibits inflammatory responses, following prevention of fibrosis, hypertrophy, and apoptosis of cardiomyocytes.

The authors described that a total of 146 DCM children (aged <18 years) were enrolled in Beijing Anzhen Hospital; however, the number of DCM children was described as 176 in Figure 1. Please confirm the number of enrolled patients.

[Response] Thanks for your suggestion. The number of DCM children enrolled in the study was 176, the text has been revised.

Abstract:

This study was a double-center, observational, prospective, integrative omics study, aiming to determine the diagnostic and prognostic value of serum sST-2 level in children with DCM. The authors have to clearly describe the outcome of this study in the abstract. I guess that the outcome will seem to the event-free survival. 

 [Response] Thanks for your comment. The outcome in our study was incident adverse outcomes during the follow-up period, which has been demonstrated clearly in the Abstract section.

Introduction:

Introduction was well written.

 [Response] Thanks for your careful review.

Methods:

There are several concerns regarding the method of this study. This study aimed to determine the association of serum level of sST-2 and beta-blocker therapy. Therefore, it is required to exactly explore the feasibility and utility of measuring serum levels of sST-2 to predict outcomes for pediatric DCM, and the effects on beta-blocker therapy, i.e. whether beta-blacker therapy could reduce serum levels of sST-2. At first, the authors have to analyze the event-free survival in overall patients, and to identify the factors relevant to the survival, including treatment with a beta-blocker and serum levels of sST-2. Subsequently, if high serum levels of sST-2 would be identified as a poor prognostic factor, the authors have to provide the ROC curve to identify a cut-off value, the sensitivity and specificity of sST-2 levels among children with DCM. Finally, the authors have to study whether beta-blocker therapy contributes to poor outcomes and could reduce serum levels of sST-2.

[Response] Thanks for your suggestion. The suggestions on the statistical analysis were valuable when we investigated the predictive value of sST2 in predicting adverse events. However, in our present study, we aimed to investigate the combined effect of sST2 and the use of β-blocker inhibitors on adverse events in PDCM, not just the single predictive value of sST2. Thus the primary analysis were performed by combining sST2 and β-blocker inhibitors. In our future study focusing on the single role of sST2, we would perform the statistical analysis as your suggestions. Thanks for your valuable suggestions again.

The authors have to describe the indication of beta-blocker therapy, and types of beta-blockers in the study population.

 [Response] Thanks for your suggestion. We did not collected detailed types of beta-blockers used in the study, which is one of the limitations of the study, and we have demonstrated this in the revised manuscript.

The authors have to describe the categories of DCM in the study population, including idiopathic, post-myocarditis, ischemic, neuromuscular, and metabolic disorders, since pediatric DCM encompasses the different pathophysiology and predisposing diseases.

 [Response] Thanks for your comment. The categories of DCM in the study population were not collected in our present study. The reason for this limitation is that the aim of the study was to investigate the effect of sST2 and β-blocker inhibitors on adverse outcomes, no matter what kinds of adverse outcomes are. Once the associations were determined, future studies would further investigate the effect of sST2 and β-blocker inhibitors on different categories of adverse outcomes with a large sample size. This limitation has been added into the revised manuscript in the discussion section.

Results:

Serum levels of sST-2, including ranges, in overall patients should be described.

 [Response] Thanks for your suggestion. The serum levels of sST2 in overall patients have been described in the revised manuscript.

The entire of results should be rewritten according to suggestions described as the above.

 [Response] Thanks for your suggestion. The suggestions on the statistical analysis were valuable when we investigated the predictive value of sST2 in predicting adverse events. However, in our present study, we aimed to investigate the combined effect of sST2 and the use of β-blocker inhibitors on adverse events in PDCM, not just the single predictive value of sST2. Thus the primary analysis were performed by combining sST2 and β-blocker inhibitors. In our future study focusing on the single role of sST2, we would perform the statistical analysis as your suggestions. Thanks for your valuable suggestions again.

Discussions

It is postulated that sST-2 is a biomarker in monitoring of heart failure, which is described in the 2017 American College of Cardiology/American Heart Association guidelines. The discussions should be strengthened to focus on the relationship between the pathophysiology of pediatric DCM and chronic inflammatory responses which deteriorates

[Response] Thanks for your suggestion. This study mainly emphasizes the effect of β-blockers on sST2, not the harmfulness of sST2, so adding the inflammatory pathological mechanism of sST2 is a bit far-fetched.

Round 2

Reviewer 3 Report

I checked it and found it appropriately collected.

Author Response

[Response] Thanks for your comment. The cutoff for sST2 was selected as the median value, which was referred to previous publishes1,2,3. As your suggested, we have performed a ROC curve (Figure R1), and the result showed the area under the curve was 0.78(95% CI, 0.66-0.89; P<0.001), the cutoff point was 25.26 with a sensitivity of 0.75 and a specificity of 0.73. In our study, the median value of sST2 was 23.76 ng/mL, which is near the cutoff value, the results were robust when we used the cutoff to define different categories.

References

  1. Hartopo AB, Sukmasari I, Puspitawati I. The Utility of Point of Care Test for Soluble ST2 in Predicting Adverse Cardiac Events during Acute Care of ST-Segment Elevation Myocardial Infarction. Cardiol Res Pract. 2018 Jun 26;2018:3048941.
  2. Choi YB, Lee MJ, Park JT, Han SH, Kang SW, Yoo TH, Kim HJ. Prognostic value of soluble ST2 and soluble LR11 on mortality and cardiovascular events in peritoneal dialysis patients. BMC Nephrol. 2020 Jun 15;21(1):228.
  3. Miftode RS, Constantinescu D, Cianga CM, Petris AO, Timpau AS, Crisan A, Costache II, Mitu O, Anton-Paduraru DT, Miftode IL, Pavel-Tanasa M, Cianga P, Serban IL. A Novel Paradigm Based on ST2 and Its Contribution towards a Multimarker Approach in the Diagnosis and Prognosis of Heart Failure: A Prospective Study during the Pandemic Storm. Life (Basel). 2021 Oct 13;11(10):1080. 
